# Effect of Lipopolysaccharide (LPS) on Oxidative Stress and Apoptosis in Immune Tissues from *Schizothorax prenanti*

**DOI:** 10.3390/ani15091298

**Published:** 2025-04-30

**Authors:** Jiqin Huang, Wei Jiang, Hongying Ma, Han Zhang, Hu Zhao, Qijun Wang, Jianlu Zhang

**Affiliations:** 1Shaanxi Key Laboratory of Qinling Ecological Security, Shaanxi Institute of Zoology, Xi’an 710032, China; huangjq1985@163.com (J.H.); jiangwei197981@163.com (W.J.); mhying7916@163.com (H.M.); hanhanr9@163.com (H.Z.); zhaohu2007@126.com (H.Z.); wqjab@126.com (Q.W.); 2College of Urban and Environmental Sciences, Northwest University, Xi’an 710127, China

**Keywords:** oxidative stress, apoptosis, lipopolysaccharide, enzymatic activities, *Schizothorax prenanti*

## Abstract

*Schizothorax prenanti* is a valuable and commercially farmed fish in China. Injecting lipopolysaccharide (LPS) into fish can simulate the reaction of the body after infection. In this study, we found that LPS infection downregulated the *catalase* and *B-cell lymphoma/Leukemia-2* expression levels, and upregulated *B-cell lymphoma/Leukemia-2*-*associated X* and *cysteine-aspartic-specific protease-3* levels in *S. prenanti*. Meanwhile, superoxide dismutase and catalase enzymatic activities were inhibited and malondialdehyde content was increased by LPS treatment. Additionally, LPS treatment induced oxidative stress (OS) damage and apoptosis in tissue sections. This study helps us further understand the effects of bacteria or LPS on the OS and apoptosis of immune tissue in fish. Furthermore, we can increase the tolerance of fish to this OS through dietary manipulation in the future.

## 1. Introduction

Oxidative stress (OS) represents an imbalance between oxidation and antioxidation in the body caused by endogenous or exogenous harmful stimuli, which leads to the overproduction and accumulation of a large number of reactive oxygen species (ROS) [1]. Once oxidative stress occurs in the body, cellular biomolecules (such as proteins, lipids, and DNA) are damaged and, ultimately, cell structure and function become impaired [2].

Apoptosis, or physiological cell death, plays a key role in the regulation of the development and maintenance of cellular homeostasis in organisms. Apoptosis, considered a cellular suicide program, is activated in cells in response to intracellular damage or physiological stimuli [3,4]. Previous studies have demonstrated that OS can induce apoptosis through intrinsic apoptotic pathways [5,6].

In intrinsic apoptotic pathways, the Bcl-2 family of proteins is a key factor. The Bcl-2 family proteins have opposing functions, i.e., inhibition and promotion of apoptosis, and the interaction between them has a regulatory effect on apoptosis [7]. Once an apoptotic signal is received, the Bcl-2 family of proteins opens their mitochondrial permeability transition pores, releasing cytochrome c into the cytosol [8]. The cytochrome c and dATP-dependent formation of Apaf-1/cysteine-aspartic-specific protease-9 (caspase-9) initiate an apoptotic protease cascade [9]. This activation ultimately triggers downstream caspase-3 activation, leading to DNA degradation in the nucleus and subsequent cell apoptosis [10,11]. The ratio of BCL2-associated X (Bax) to Bcl-2 is a “molecular switch” that determines the response of a cell to apoptotic stimuli [12]. Bax and Bcl-2 regulate apoptosis by forming homologous or heterodimers. Homologous dimer formed by Bax induces apoptosis, while the formation of a heterodimer between Bax and Bcl-2 inhibits apoptosis. The Bax/Bcl-2 ratio also determines the degree of opening of various channels in the outer mitochondrial membrane, forming a hub for cell apoptosis regulation. Therefore, the cell fate ultimately depends on the balance between Bcl-2 and Bax proteins [13,14].

Lipopolysaccharide (LPS) is the main component of the outer membrane of Gram-negative bacteria. Endotoxins are named after the biological activities induced by LPS when they enter the host body [15]. Previous studies have demonstrated that LPS induces OS and apoptosis in different fish tissues. By detecting the enzymatic activities of CAT and SOD, the malondialdehyde (MDA) content, and the expression of apoptosis-related genes, Li et al. [16] found that LPS could induce OS and apoptosis in the intestines and hepatopancreas of the common carp (*Cyprinus carpio*). LPS can also stimulate ROS production and cysteinyl aspartate-specific proteinase-3 (caspase-3) activity in fin cell lines from red crucian carp (*Carassius auratus* red var.), white crucian carp (*Carassius cuvieri*), and their hybrid offspring [17]. However, LPS did not seem to cause OS in the muscle cells of the Yellow River carp (*Cyprinus carpio haematopterus*) nor did it affect the expression of CAT and SOD [18].

As a stressor, this study aimed to explore the tissue differences in *S. prenanti* in response to LPS-induced OS and apoptosis. Histological changes and apoptosis were evaluated in the liver, head kidney (HK), and spleen. The mRNA expression of *CAT*, *Bcl-2*, *Bax*, and *caspase-3* was analyzed, and the total SOD (T-SOD) and CAT enzymatic activities and MDA content were measured. Our findings are expected to contribute to a better understanding of the responses of different tissues to bacterial challenges.

## 2. Materials and Methods

### 2.1. Experimental Fish

The experimental *S. prenanti* (109.3 ± 27.1 g) were selected from a cold-water fish farm in Hanzhong City, China. Before the experiment, fish were kept in glass tanks (60 × 40 × 30 cm^3^), with aerated water at a temperature of 19 ± 1 °C. The experimental fish were cultured as reported in our previous study [19]. The animal study protocol was approved by the Animal Ethics Committee of the Shaanxi Institute of Zoology (Protocol code: L23D003A55, date of approval: 5 January 2023) for studies involving *S. prenanti*.

### 2.2. LPS Stimulation and Sampling

After 12 d of acclimatization, 36 similarly sized *S. prenanti* fish were selected and divided into two groups. In the control group, the fish were stimulated with phosphate-buffered saline (PBS), while the fish in the test group were intraperitoneally inoculated with LPS (L2880, Sigma, St. Louis, MO, USA) at a dose of 10 mg/kg of body weight. Six individuals were randomly sampled from each group at 0, 12, and 24 h post injection. Fish were euthanized with 100 mg/L of tricaine methanesulfonate (MS-222; Sigma, USA) and the liver, HK, and spleen were collected, frozen in liquid nitrogen, and stored at −80 °C until further gene expression and enzymatic activity analyses. The liver, HK, and spleen were fixed in 4% paraformaldehyde.

### 2.3. Effect of LPS on CAT, Bcl-2, Bax, and Caspase-3 Expression

Total RNA was isolated from the liver, HK, and spleen tissues using TRIzol reagent (Invitrogen, Carlsbad, CA, USA). Total RNA purity and concentration were measured using agarose gel electrophoresis and spectrophotometry, respectively. cDNA was synthesized using the RevertAid First Strand cDNA Synthesis Kit (Thermo Scientific, Vilnius, Lithuania), according to the manufacturer’s instructions. Real-time quantitative PCR (qPCR) was performed in a total volume of 20 μL, containing 10 μL of FastStart Essential DNA Green Master (Roche Diagnostics, Risch-Rotkreuz, Switzerland), 0.5 μL each of the forward and reverse primers (10 μmol/mL), 1 μL of cDNA, and 8 μL of ddH_2_O. PCR amplification was performed using Applied Biosystems StepOnePlus (Life Technologies, Carlsbad, CA, USA). *β-actin* was used as the reference gene. Every sample was amplified in triplicate. After each run, melting curves for each gene were constructed to confirm the specificity of the primers. The data were analyzed according to the 2^−ΔΔCT^ method [20] and normalized to the mean expression of *β-actin*. The primers used in this study are listed in Table 1.

### 2.4. Assay for Determining Antioxidant Status

The liver, HK, and spleen samples were homogenized, and T-SOD and CAT activities and MDA content in the homogenate were measured using SOD, CAT, and MDA assay kits (Jiancheng Biotech. Co., Nanjing, China), respectively. Total protein concentration was measured using the Total Protein Assay Kit (Jiancheng Biotech. Co., Nanjing, China; based on standard BCA method).

### 2.5. Hematoxylin–Eosin Detection 

The liver, HK, and spleen samples were fixed in 4% paraformaldehyde overnight at 4 °C, and then washed and dehydrated with a gradient ethanol solution, cleared using xylene, and embedded in paraffin. Thereafter, 6 μm thick sections were prepared. After dewaxing and hydration, the tissue sections were stained with hematoxylin–eosin (HE) and observed under a light microscope (Olympus, Tokyo, Japan).

### 2.6. Detection of Apoptotic Cells

Tissue sections obtained from the liver, HK, and spleen samples were dewaxed and hydrated. Apoptotic cells were stained using the In Situ Cell Death Detection Kit, POD (Roche, Raleigh, NC, USA), according to the manufacturer’s protocol. Briefly, the slides were incubated with 20 µg/mL of proteinase K (TIANGEN, Beijing, China) at room temperature. Endogenous peroxidase was inactivated with 2% H_2_O_2_, and each slide was blocked with 5% BSA (BioFROXX, Hefei, China). The slides were then labeled with a mixture of terminal deoxynucleotidyl transferase (TdT) and biotinylated dUTP buffer. Finally, the samples were stained with DAPI. The slides were photographed using Pannoramic MIDI (3DHistech, Budapest, Hungary).

### 2.7. Statistical Analysis

Data are presented as the mean ± standard deviation. Data distribution was verified using the Shapiro–Wilk test, which showed that all the data had an appropriate normal distribution. The data of qPCR and enzymatic activity were compared using two-way repeated-measures analysis of variance (ANOVA) to evaluate the effects of LPS and time followed by post hoc Bonferroni’s test. A significance level of 0.05 was used in all tests (*p* < 0.05).

## 3. Results

### 3.1. Expression of CAT, Bax, Bcl-2, and Caspase-3 After LPS Administration

qPCR was conducted to quantify the changes in mRNA levels of *CAT*, *Bax*, *Bcl-2*, and *caspase-3* in the liver, HK, and spleen tissues at 0 h, 12 h, and 24 h after LPS administration (Figure 1).

#### 3.1.1. Expression of *CAT*

After LPS injection into the liver, there was a significant main effect of LPS (*p* = 0.0004), significant main effect of time (*p* < 0.0001), and a significant interaction for LPS × time (*p* < 0.0004) (Figure 1a). This result revealed that LPS significantly reduced *CAT* expression at 12 h (*p* < 0.001) and 24 h (*p* < 0.001).

After HK infection with LPS, there was a significant main effect of LPS (*p* < 0.0001), significant main effect of time (*p* < 0.0001), and a significant interaction for LPS × time (*p* < 0.0001) (Figure 1b). The *CAT* expression was significantly attenuated in the LPS group at 12 h (*p* < 0.001) and 24 h (*p* < 0.001).

In the spleen with LPS administration, there was a significant main effect of LPS (*p* = 0.0001), significant main effect of time (*p* < 0.0001), and a significant interaction for LPS × time (*p* < 0.0001) (Figure 1c). We found that LPS significantly decreased *CAT* expression at 12 h (*p* < 0.001) and 24 h (*p* < 0.001).

#### 3.1.2. Expression of *Bax*

After liver infection with LPS, there was a significant main effect of LPS (*p* < 0.0001), significant main effect of time (*p* < 0.0001), and a significant interaction for LPS × time (*p* < 0.0001) (Figure 1d). We found that LPS significantly increased *Bax* expression at 24 h (*p* < 0.001). In addition, there was a cumulative effect in *Bax* expression with the extension of time.

After HK administration with LPS, there was a significant main effect of LPS (*p* < 0.0001), significant main effect of time (*p* < 0.0001), and a significant interaction for LPS × time (*p* < 0.0001) (Figure 1e). The qPCR analysis revealed that LPS significantly elevated *Bax* expression at 12 h (*p* < 0.001) and 24 h (*p* < 0.001). Moreover, there was a cumulative effect in *Bax* expression with the extension of time.

After LPS injection into the spleen, there was a significant main effect of LPS (*p* < 0.0001), significant main effect of time (*p* < 0.0001), and a significant interaction for LPS × time (*p* < 0.0001) (Figure 1f). We found that LPS significantly increased *Bax* expression at 12 h (*p* < 0.001) and 24 h (*p* < 0.001). And this *Bax* expression exhibited a time-related upward trend.

#### 3.1.3. Expression of *Bcl-2*

After LPS injection into the liver, there was a significant main effect of LPS (*p* = 0.0071), significant main effect of time (*p* < 0.0001), and a significant interaction for LPS × time (*p* < 0.0001) (Figure 1g). We found that LPS significantly decreased *Bcl-2* expression at 12 h (*p* < 0.05) and 24 h (*p* < 0.001).

After HK infection with LPS, there was a significant main effect of LPS (*p* = 0.0492), significant main effect of time (*p* = 0.0075), and a significant interaction for LPS × time (*p* = 0.0279) (Figure 1h). The result revealed that *Bcl-2* expression was significantly attenuated in the LPS group at 24 h (*p* < 0.01).

In the spleen with LPS stimulation, there was a significant main effect of LPS (*p* = 0.0001), significant main effect of time (*p* = 0.0013), and a significant interaction for LPS × time (*p* = 0.0005) (Figure 1i). This result suggested that LPS significantly decreased *Bcl-2* expression at 12 h (*p* < 0.01) and 24 h (*p* < 0.001).

#### 3.1.4. Expression of *Caspase-3*

After liver infection with LPS, there was a significant main effect of LPS (*p* < 0.0001), significant main effect of time (*p* < 0.0001), and a significant interaction for LPS × time (*p* < 0.0001) (Figure 1j). We found that LPS significantly increased *caspase-3* expression at 12 h (*p* < 0.001) and 24 h (*p* < 0.001). In addition, the *caspase-3* expression exhibited a time-related upward trend.

After LPS injection into the HK, there was a significant main effect of LPS (*p* < 0.0001), significant main effect of time (*p* < 0.0001), and a significant interaction for LPS × time (*p* < 0.0001) (Figure 1k). This result suggested that LPS significantly increased *caspase-3* expression at 12 h (*p* < 0.001) and 24 h (*p* < 0.001). With the extension of time, the *caspase-3* expression followed an upward trend.

After spleen infection with LPS, there was a significant main effect of LPS (*p* < 0.0001), significant main effect of time (*p* < 0.0001), and a significant interaction for LPS × time (*p* < 0.0001) (Figure 1l). We found that LPS significantly increased *caspase-3* expression at 12 h (*p* < 0.001) and 24 h (*p* < 0.001). Moreover, there was a cumulative effect in *caspase-3* expression with the extension of time.

### 3.2. Antioxidant Status Under LPS-Induced Stress

To evaluate the effects of LPS and time on the OS status in *S. prenanti*, T-SOD and CAT activities and the MDA content in the liver, HK, and spleen tissues were measured.

#### 3.2.1. T-SOD Activity

After LPS injection into the liver, there was a significant main effect of LPS (*p* = 0.0038), with no significant effect of time and no significant interaction for LPS × time (Figure 2a). We found that LPS significantly decreased SOD activity at 12 h (*p* < 0.001) and 24 h (*p* < 0.05).

After HK infection with LPS, there was no significant effect of LPS, with no significant effect of time and no significant interaction for LPS × time (Figure 2b). This result revealed that LPS had no significant effect on SOD activity at all times.

In the spleen with LPS stimulation, there was a significant main effect of LPS (*p* = 0.0036), with no significant effect of time and no significant interaction for LPS × time (Figure 2c). We found that LPS significantly decreased SOD activity at 12 h (*p* < 0.01).

#### 3.2.2. CAT Activity

After liver infection with LPS, there was a significant main effect of LPS (*p* = 0.0111) and a significant interaction for LPS × time (*p* = 0.0092), with no significant effect of time (Figure 2d). We found that LPS significantly decreased CAT activity at 12 h (*p* < 0.01) and 24 h (*p* < 0.05).

After LPS injection into the HK, there was a significant main effect of LPS (*p* = 0.0022), significant main effect of time (*p* = 0.0011), and a significant interaction for LPS × time (*p* = 0.008) (Figure 2e). This result suggested that LPS significantly decreased CAT activity at 12 h (*p* < 0.001) and 24 h (*p* < 0.01).

After spleen stimulation with LPS, there was a significant main effect of LPS (*p* = 0.0013), significant main effect of time (*p* = 0.0062) and a significant interaction for LPS × time (*p* = 0.0176) (Figure 2f). The LPS significantly decreased CAT activity at 12 h (*p* < 0.001) and 24 h (*p* < 0.01).

#### 3.2.3. MDA Content

After LPS injection into the liver, there was a significant main effect of LPS (*p* = 0.0083), significant main effect of time (*p* = 0.0383), and a significant interaction for LPS × time (*p* = 0.0079) (Figure 2g). This result suggested that LPS significantly increased MDA content at 12 h (*p* < 0.001) and 24 h (*p* < 0.05).

After LPS induction in HK, there was a significant main effect of LPS (*p* = 0.0193), significant main effect of time (*p* = 0.0068), and a significant interaction for LPS × time (*p* = 0.0182) (Figure 2h). The result revealed that LPS significantly increased MDA content at 12 h (*p* < 0.01).

In the spleen with LPS administration, there was a significant main effect of LPS (*p* = 0.0046), significant main effect of time (*p* < 0.0001), and a significant interaction for LPS × time (*p* < 0.0001) (Figure 2i). We found that LPS significantly increased MDA content at 12 h (*p* < 0.001) and 24 h (*p* < 0.05).

### 3.3. Histopathological Analysis

To observe the effects of LPS on the liver, HK, and spleen, we performed HE staining. Figure 3 illustrates the histological changes in the three tissues 12 and 24 h after LPS injection. We observed enlarged nuclei in hepatocytes and atrophy of the hepatic cell membrane after LPS stimulation for 12 and 24 h (Figure 3b,c). In HK, necrotic exfoliated epithelial cells and cell fragments were observed in the lumen of the renal tubules and renal tubule epithelial cells (Figure 3e,f). Notably, large areas of hemosiderin deposits were observed in the spleen (Figure 3h,i). LPS affected the liver and spleen more than the HK. Inflammatory injury was observed in all three tissues. In the case of inflammatory responses, the nucleus may enlarge because of the stimulation of DNA synthesis and repair within the cell.

### 3.4. Induction of Apoptosis by LPS

The liver, HK, and spleen tissues were collected for terminal deoxynucleotidyl transferase (TdT)-mediated dUTP nick-end labeling (TUNEL) staining. Compared to the control group, LPS treatment led to an increase in the number of apoptotic cells, especially in the liver (Figure 4).

## 4. Discussion

In recent years, intensive feeding of *S. prenanti* has resulted in a series of environmental stressors [21,22]. A range of physiological and biochemical reactions are triggered in fish in response to various stressors, including antioxidant, inflammatory, and immune responses [23]. A previous study demonstrated that LPS increased the expression of heat shock proteins [19]. In this study, we evaluated the effects of LPS and time on OS in the liver, HK, and spleen of *S. prenanti* and the differences in response to OS among these three tissues.

CAT, T-SOD, and MDA levels are basic indices for evaluating the antioxidant capacity [24]. In general, ROS production is limited in organisms and ROS is scavenged by SOD, CAT, and GSH-Px [25]. However, external stimulation induces the production of large amounts of ROS, leading to lipid peroxidation. An increase in the MDA content, which is the end product of lipid peroxidation, indicates cellular damage [26]. In the current study, LPS stimulation notably increased MDA levels in the liver, HK, and spleen at 12 h and (or) 24 h, providing evidence for LPS-induced oxidative damage in these three tissues of *S. prenanti*. These results are consistent with those reported by Mohamadin et al. [27] and Gu et al. [28]. Our results indicate that the liver, which is a critical organ for xenobiotic metabolism and detoxification [29,30], is highly sensitive to OS. In addition, reduced CAT activity and mRNA expression after 12 h of LPS exposure may contribute to mRNA turnover [31] to maintain or increase ROS levels in host cells. Elevated ROS levels promote pathogen elimination via direct oxidative damage or through various innate and adaptive mechanisms [32]. SOD plays an important role in OS defense [33]. In the present study, decreased SOD activity was observed in LPS-treated *S. prenanti*, which is consistent with findings in the Chinese mitten crab [34]. In the histopathology analysis, LPS was found to have aggravated the damage to the liver, HK, and spleen to varying degrees, such as structural damage and hemolytic aggregation, which is consistent with the findings of Liu et al. [34]. These results indicate that LPS disrupts the balance of the REDOX system and causes oxidative damage. OS has been reported to disrupt the homeostatic balance in the liver, resulting in tissue damage, injury, and remodeling [35], which may explain why T-SOD and CAT enzymatic activities were elevated 24 h after LPS administration.

Studies have shown that LPS induces apoptosis in fish leukocytes [36,37]. Bcl-2 family members (Bcl-2 and Bax) and caspase family members (caspase-3, caspase-8, and caspase-9) play vital roles in the regulation of apoptosis [38,39]. In the current study, LPS administration significantly increased the mRNA levels of *Bax* and *caspase-3* and decreased those of *Bcl-2*. TUNEL analysis showed that the number of apoptotic cells significantly increased after LPS stimulation for 12 and 24 h. These results indicate that LPS stimulation enhanced apoptosis in *S. prenanti*. These results are similar to those reported for the Chinese mitten crab [34], brook trout [40], and Atlantic salmon [41].

Histological changes provide a method to visualize and characterize the effects of LPS stimulation in vivo. Immune organs such as the liver, kidney, and spleen are particularly sensitive to pathogens. It has been reported that Gram-negative bacteria induce an inflammatory response and pathological changes in the immune tissues of fish [42,43,44], which is consistent with our results. Notably, LPS administration to *S. prenanti* resulted in splenic inflammation and hemosiderin deposition. Studies have demonstrated that impaired iron utilization in the body can lead to the accumulation of hemosiderin in the spleen or significant destruction of red blood cells, which serve as important indicators of environmental stress in fish [45]. Therefore, the detection of splenic hemosiderin is important for monitoring the health of *S. prenanti*. In summary, the pathological changes and abnormalities in the liver, HK, and spleen of *S. prenanti* in a physiological state can aid in the diagnosis of various diseases.

## 5. Conclusions

To our knowledge, this is the first study to report the effects of LPS stimulation on OS and apoptosis in the liver, HK, and spleen of *S. prenanti*. The decrease in CAT and SOD enzymatic activity and/or gene expression and the increase in MDA content indicated that LPS did induce OS. Further, OS induced apoptosis through the intrinsic apoptotic pathway in *S. prenanti*. In view of the tissue-specific expression of apoptosis-related genes, the mechanism of apoptosis in *S. prenanti* needs further study.

## Figures and Tables

**Figure 1 animals-15-01298-f001:**
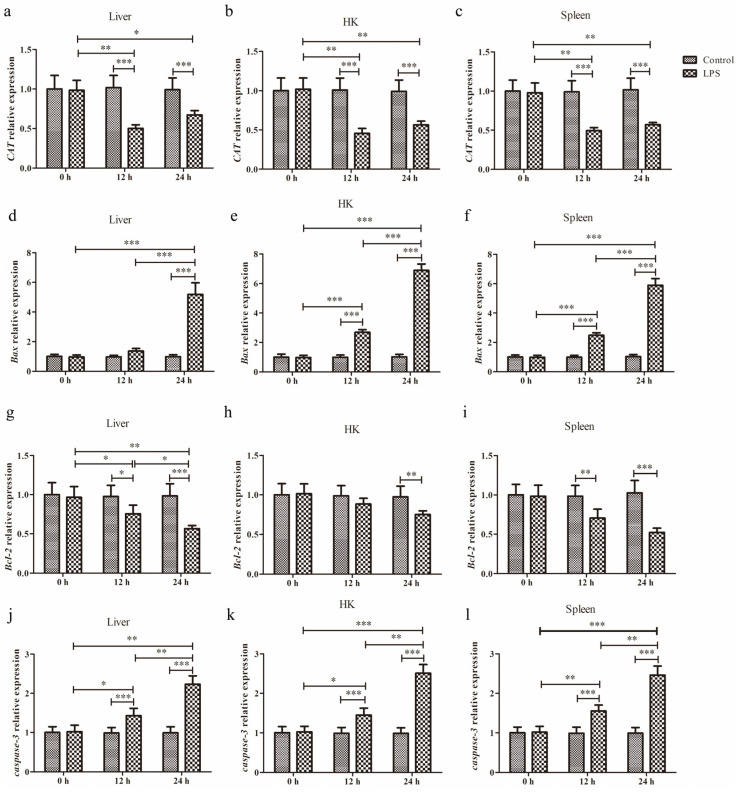
The relative expression levels of *CAT*, *Bax*, *Bcl-2*, and *caspase-3* in the liver, HK, and spleen of *Schizothorax prenanti* at 0, 12, and 24 h post-LPS injection. All the expression data were calculated by normalizing to the *β-actin* reference gene and were then shown in a histogram by comparing to the expression level of the control at 0 h (fold change of 1.0). (**a**–**c**) *CAT* expression levels; (**d**–**f**) *Bax* expression levels; (**g**–**i**) *Bcl-2* expression levels; (**j**–**l**) *caspase-3* expression levels. Data are presented as mean ± SD (*n* = 6). Significant differences between the groups are marked with asterisks (* *p* < 0.05, ** *p* < 0.01, and *** *p* < 0.001). *CAT*, catalase; HK, head kidney; LPS, lipopolysaccharide.

**Figure 2 animals-15-01298-f002:**
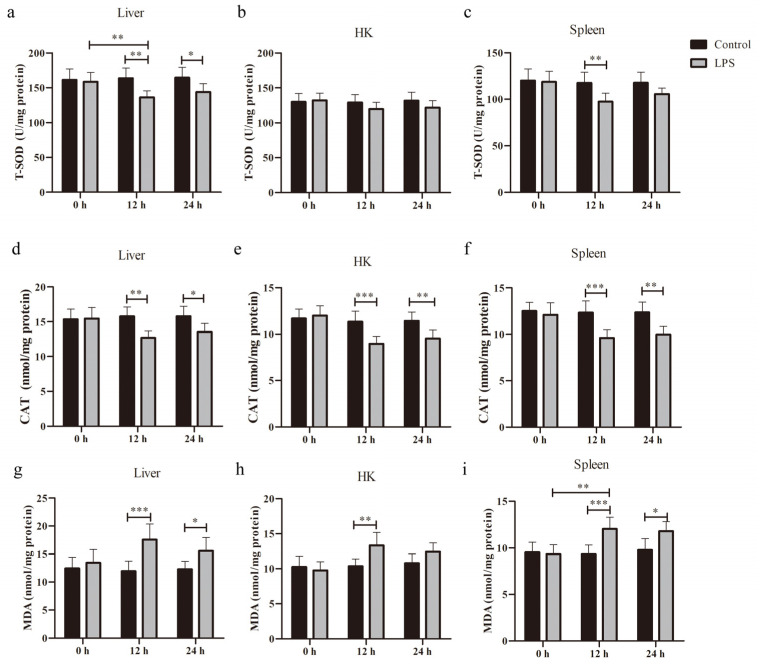
Oxidation indices in the liver, HK, and spleen from *Schizothorax prenanti* at 0, 12, and 24 h post-LPS injection. (**a**–**c**) T-SOD activity; (**d**–**f**) CAT activity; (**g**–**i**) MDA content. Significant differences between the groups are marked with asterisks (* *p* < 0.05, ** *p* < 0.01, and *** *p* < 0.001). Data are presented as mean ± SD (*n* = 6). T-SOD, total superoxide dismutase; CAT, catalase; MDA, malondialdehyde; HK, head kidney; LPS, lipopolysaccharide.

**Figure 3 animals-15-01298-f003:**
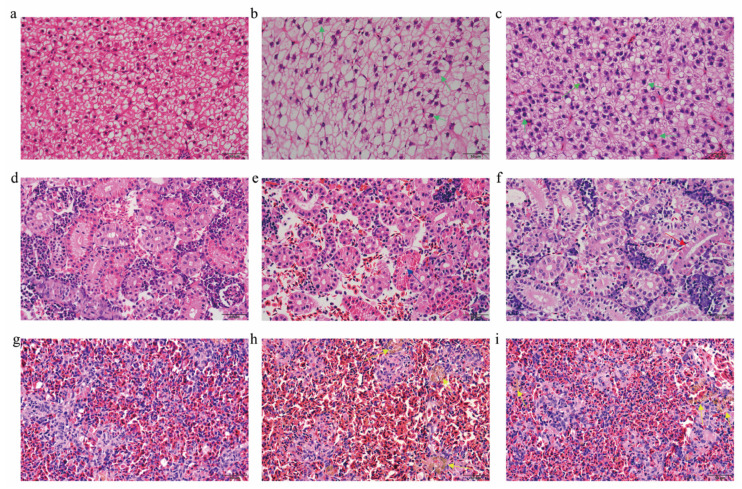
The effect of LPS on the histology of the liver, HK, and spleen from *Schizothorax prenanti*. (**a**,**d**,**g**) The liver, HK, and spleen tissues of the control group; (**b**,**e**,**h**) the liver, HK, and spleen tissues of LPS-injected fish after 12 h of treatment; and (**c**,**f**,**i**) the liver, HK, and spleen tissues of LPS-injected fish after 24 h of treatment. The green arrow indicates enlarged nuclei; the blue arrow indicates necrotic, exfoliated epithelial cells and cell fragments in the lumen of the renal tubules; the red arrow indicates exfoliated renal tubular cells; and the yellow arrow indicates hemosiderin deposits. Scale bar = 50 μm. HK, head kidney; LPS, lipopolysaccharide.

**Figure 4 animals-15-01298-f004:**
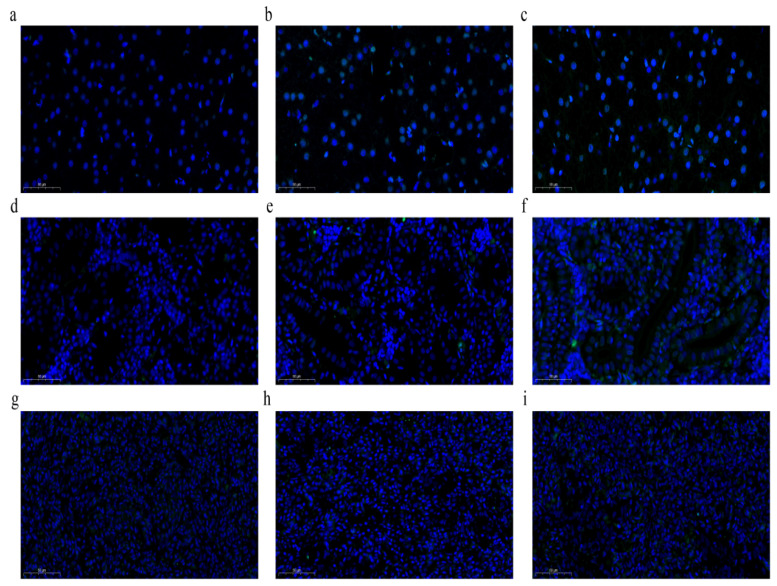
The effect of LPS on cellular apoptosis in the liver, HK, and spleen of *Schizothorax prenanti*. (**a**,**d**,**g**) The liver, HK, and spleen tissues of the control group; (**b**,**e**,**h**) the liver, HK, and spleen tissues of LPS-injected fish after 12 h of treatment; and (**c**,**f**,**i**) the liver, HK, and spleen tissues of LPS-injected fish after 24 h of treatment. Scale bar = 50 μm. HK, head kidney; LPS, lipopolysaccharide.

**Table 1 animals-15-01298-t001:** The primers used for real-time quantitative PCR.

Primer	Accession Number	Sequence (5′-3′)	Annealing Temperature (°C)	PCR Efficiency	Size (bp)
Bcl-2	OQ734947	F: CTGGATGACAGACTACCTGAAC	62	97.5%	118
		R: CGACAATGGGTGGAACATAGA			
Bax	OQ347970	F: GACTCCACTCTTCAACCAACTC	62	98.1%	116
		R: AGCCGACATGCAAAGTAGAA			
CAT	OQ737946	F: GGAAACAACACTCCCATCTT	62	101.5%	120
		R: CCAGAAGTCCCAAACCATATC			
caspase-3	OQ737945	F: CAGTCACATGCCTTCAGATAC	62	103.5%	122
		R: GCATCTACATCAGTACCATTCC			
β-actin	MK439425	F: GACCACCTTCAACTCCATCAT	62	99.5%	126
		R: GTGATCTCCTTCTGCATCCTATC			

## Data Availability

Publicly available datasets were analyzed in this study. The rest of the data presented in this study are available on request from the corresponding author.

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
