# Peer review of "Effect of Lipopolysaccharide (LPS) on Oxidative Stress and Apoptosis in Immune Tissues from Schizothorax prenanti"

_animals, 2025, doi:10.3390/ani15091298_

Round 1

Reviewer 1 Report

Comments and Suggestions for Authors

a. The introduction is thorough, giving a solid rundown of both the general mechanisms behind LPS-induced stress and studies focused specifically on fish. With references stretching up to 2023, it’s clear the work stays current and relevant.

b. The experimental design suits the goal of digging into tissue-specific responses to LPS stress. That said, one thing could use a bit more clarity: the "0 hours post-injection" sampling. The text hints it’s right after the injection, but for a true baseline, it’d be worth spelling out whether this is pre-treatment or post-treatment. If it’s immediately after, it might not fully catch the fish’s starting point before stress kicks in.

c. The methods section is detailed enough that someone could repeat the study, and listing the kits used is a nice touch.

Still, a few tweaks could make it even stronger:

  1. Pin down the exact number of fish per group and time point—say, 6 fish each time, based on the total of 36 and the sampling schedule.
  2.  For qPCR, toss in some details on controls, like doing a melt curve analysis to confirm the primers are hitting the right target.
  3. For the histology and TUNEL work, spell out how apoptotic cells were counted—maybe the standards used or the number of fields checked.

These small additions would really tighten up the reproducibility, especially for folks working in fish immunology.

d. The results come across clearly, with stats showing significance (p<0.05 to p<0.001) and solid backing from figures and tables. No big overhaul needed here, though making sure every figure has clear labels and gets a proper shout-out in the text would smooth things out even more. Highlighting the size of the changes—like fold shifts in gene expression—helps put the findings in context.
e. The conclusions lay it out plainly: LPS sets off oxidative stress and apoptosis through the mitochondrial pathway, hitting the liver, head kidney, and spleen in S. prenanti.

This is the first study to map these effects in this species, and there’s no disconnect with the data—it all holds up.

No tweaks needed; it’s a strong wrap-up that adds real insight into how fish handle bacterial infections in aquaculture.

Author Response

Reviewer 1

Comments a. The introduction is thorough, giving a solid rundown of both the general mechanisms behind LPS-induced stress and studies focused specifically on fish. With references stretching up to 2023, it’s clear the work stays current and relevant.

Response a: Thank you for your recognition of our work.

  1. The experimental design suits the goal of digging into tissue-specific responses to LPS stress. That said, one thing could use a bit more clarity: the "0 hours post-injection" sampling. The text hints it’s right after the injection, but for a true baseline, it’d be worth spelling out whether this is pre-treatment or post-treatment. If it’s immediately after, it might not fully catch the fish’s starting point before stress kicks in.

Response b: Thank you for pointing this out. The reason why we chose to sample 0 hours post-injection is that injection with a syringe will also cause stress to the body. In order to offset this stress, we chose to sample immediately after injection. You’re right that LPS had no effect at 0 hours. If you still find it misleading, we will remove this group later.

  1. The methods section is detailed enough that someone could repeat the study, and listing the kits used is a nice touch.

Still, a few tweaks could make it even stronger:

  1. Pin down the exact number of fish per group and time point—say, 6 fish each time, based on the total of 36 and the sampling schedule.
  2. For qPCR, toss in some details on controls, like doing a melt curve analysis to confirm the primers are hitting the right target.
  3. For the histology and TUNEL work, spell out how apoptotic cells were counted—maybe the standards used or the number of fields checked.

These small additions would really tighten up the reproducibility, especially for folks working in fish immunology.

Response c: Thank you for pointing this out.

About the exact number of fish per group and time point were written in page 3 line 94 and line 97.

The melt curve analysis was performed in this study, according to your advise, we have added this content in page 3 line 113-114.

We randomly selected 20 visual fields to count the number of apoptotic cells. After confirming that LPS could cause apoptosis, we chose a representative picture and did not perform statistical analysis of apoptotic cells because the number of apoptotic cells could not be accurately recorded like flow cytometry.

  1. The results come across clearly, with stats showing significance (p<0.05 to p<0.001) and solid backing from figures and tables. No big overhaul needed here, though making sure every figure has clear labels and gets a proper shout-out in the text would smooth things out even more. Highlighting the size of the changes—like fold shifts in gene expression—helps put the findings in context.

Response d: Thank you for your recognition of our work. According to your advise, we have added the size of the changes including gene expression in page 4 line 158, page 4 line 167-168 and page 5 line 182-183.

  1. The conclusions lay it out plainly: LPS sets off oxidative stress and apoptosis through the mitochondrial pathway, hitting the liver, head kidney, and spleen in S. prenanti.

This is the first study to map these effects in this species, and there’s no disconnect with the data—it all holds up.

No tweaks needed; it’s a strong wrap-up that adds real insight into how fish handle bacterial infections in aquaculture.

Response e: Thank you for your recognition of our work.

Reviewer 2 Report

Comments and Suggestions for Authors

The manuscript is excellent, full of new and very interesting information. However, the authors were economical in certain sections of the manuscript, which are important to be complete.
I would like to begin by complimenting the authors for the quality of the tables and figures (images and graphs); clearly demonstrating the results. Congratulations.
Now, what needs to be improved:
1) In the introduction: The scientific problem that exists, as I understand it, would be oxidative stress; if that is the case, it needs to be better described what oxidative stress really is; how it relates to cell damage and even apoptosis; as well as why it increases LPS. Restructure
2) In the introduction: is oxidative stress a problem with the fish used in the tests??? At the end of the introduction, they wrote a paragraph about the fish; which was lost. Isn't the fish just an experimental model in this research?? If so, that paragraph is unnecessary; but if it is the direct target of the research, it needs to be included more clearly in the problem.

3) Statistical analysis section: rewrite; detail the previous analyses, give all the details of this analysis; for example: a) did this data have a normal distribution? was there no data transformation? (it is difficult to have normal data in expression analyses); why did you choose a parametric test?

4) Conclusion: terrible and generalist that makes no sense at all; not concluding with the excellent results that you have is a crime; authors, I invite you to rewrite; responding to the objective; in a timely and connected manner; because information further reinforces the final conclusion.

Comments on the Quality of English Language

I had no problems reading. Good

Author Response

Reviewer 2

The manuscript is excellent, full of new and very interesting information. However, the authors were economical in certain sections of the manuscript, which are important to be complete.
I would like to begin by complimenting the authors for the quality of the tables and figures (images and graphs); clearly demonstrating the results. Congratulations.
Now, what needs to be improved:
1) In the introduction: The scientific problem that exists, as I understand it, would be oxidative stress; if that is the case, it needs to be better described what oxidative stress really is; how it relates to cell damage and even apoptosis; as well as why it increases LPS. Restructure

Response 1: Thanks for your advise. We have, accordingly, restructured these contents in page 2 line 41-50.

2) In the introduction: is oxidative stress a problem with the fish used in the tests??? At the end of the introduction, they wrote a paragraph about the fish; which was lost. Isn't the fish just an experimental model in this research?? If so, that paragraph is unnecessary; but if it is the direct target of the research, it needs to be included more clearly in the problem.

Response 2: Thank you for pointing this out. We agree with this comment. The fish is an experimental model in this study. According to your advise, we have deleted this paragraph in  introduction section.

  • Statistical analysis section: rewrite; detail the previous analyses, give all the details of this analysis; for example: a) did this data have a normal distribution? was there no data transformation? (it is difficult to have normal data in expression analyses); why did you choose a parametric test?

Response 3: Thanks for your advise. We have described it in detail in the Statistical analysis section in page 4 line 146-150. And we have reanalyzed the data. Besides, The real-time quantitative data are based on relative quantitative analysis (2−ΔΔCT method) in page 3 line 114-116. The enzyme activity data is the OD value which were measured by the instrument and then converted by the formula.

  • Conclusion: terrible and generalist that makes no sense at all; not concluding with the excellent results that you have is a crime; authors, I invite you to rewrite; responding to the objective; in a timely and connected manner; because information further reinforces the final conclusion.

Response 4: Thanks for your suggestion. We have rewritten the conclusion in page 10 line 307-311.

Reviewer 3 Report

Comments and Suggestions for Authors

This study shows LPS induces oxidative stress and apoptosis in different organs of Schizothorax prenanti. It is interesting and can be published after revisions.

Simple summary: it is OK, just lacks a practical use of the present results. For example, the authors can highlight the potential benefits of boosting fish antioxidant defense (for example through dietary manipulation) to increase fish tolerance of the bacterial diseases.

Abstract: The authors should add details of LPS administration and sampling time. Also, as mentioned above, a conclusion with proposed use of the results are necessary.

Introduction: One of the route of apoptosis (also necrosis) is oxidative stress and accumulation of ROS. The authors should state it in the introduction and join it with the text they previously wrote.

Methods: There is a big concern about the LPS used! The authors stated that the LPS was purchased from Sigma Co. As far as I know, the LPS Sigma supplied is from E. coli! This bacterium is not a pathogen in fish and causes no disease!! So, why did the authors not used a LPS from fish pathogen?! The fish respond differently to LPS derived from different bacteria. At least the authors must provide evidence showing responses to E. coli LPS is similar to the pathogenic bacteria LPS. This needs discussion.

Accession number of the primers must be added.

If the authors used ANOVA, they should confirm all data were normally-distributed and homoscedastic. Moreover, as the authors sampled the fish across the time, they should analyze the results with "repeated measure two-way ANOVA". Also, the authors did not stated how they performed pair comparisons (post hoc test).

Discussion: as changing the statistical methods affects the results, I cannot review the discussion section at this round.

Author Response

Reviewer 3

This study shows LPS induces oxidative stress and apoptosis in different organs of Schizothorax prenanti. It is interesting and can be published after revisions.

Simple summary: it is OK, just lacks a practical use of the present results. For example, the authors can highlight the potential benefits of boosting fish antioxidant defense (for example through dietary manipulation) to increase fish tolerance of the bacterial diseases.

Response : Thanks very much. According to your suggestion, we have added the highlight in page 1 line 17-18.

Abstract: The authors should add details of LPS administration and sampling time. Also, as mentioned above, a conclusion with proposed use of the results are necessary.

Response : Thanks for your advise. We have added these details in page 1 line 22-23 and line 34-36.

Introduction: One of the route of apoptosis (also necrosis) is oxidative stress and accumulation of ROS. The authors should state it in the introduction and join it with the text they previously wrote.

Response : Thanks for your advise. We have added these details in page 2 line 41-50.

Methods: There is a big concern about the LPS used! The authors stated that the LPS was purchased from Sigma Co. As far as I know, the LPS Sigma supplied is from E. coli! This bacterium is not a pathogen in fish and causes no disease!! So, why did the authors not used a LPS from fish pathogen?! The fish respond differently to LPS derived from different bacteria. At least the authors must provide evidence showing responses to E. coli LPS is similar to the pathogenic bacteria LPS. This needs discussion.

Response : Thank you for pointing this out. In this study, In this study, LPS is a stressor. We found that LPS can cause oxidative stress in many kinds of fish by reviewing domestic and foreign literature. You’re right. The responses of grass carp to E. coli LPS and aeromonas hydrophila LPS were different. In order to avoid misunderstanding, we have added “ As a stressor” was in page 2 line 79, and we have deleted the content about “bacterial diseases” and “LPS is the main component of membranes of Gram-negative bacteria” in simple summary. Thanks again.

Accession number of the primers must be added.

Response: Thanks for pointing this out. We have added the accession number in Table 1.

If the authors used ANOVA, they should confirm all data were normally-distributed and homoscedastic. Moreover, as the authors sampled the fish across the time, they should analyze the results with "repeated measure two-way ANOVA". Also, the authors did not stated how they performed pair comparisons (post hoc test).

Response: Thank you for pointing this out. We are sorry for that. We have described it in detail in the Statistical analysis section in page 4 line 146-150. And we have reanalyzed the data.

Discussion: as changing the statistical methods affects the results, I cannot review the discussion section at this round.

Response: We are sorry for the trouble. In our study, the only variable was whether LPS was added or not. We looked through some literature, the groups similar to ours used one-way ANOVA, so we still used one-way ANOVA. Our knowledge of statistics is weak. Please correct us if we make any mistakes.

Round 2

Reviewer 2 Report

Comments and Suggestions for Authors

Congratulations to the authors, the paper is ready to be published. All my questions have been checked, adjusted and I agree.

Comments on the Quality of English Language

good

Author Response

Reviewer 2

Congratulations to the authors, the paper is ready to be published. All my questions have been checked, adjusted and I agree.

Response: Thank you for your recognition of our work.

Reviewer 3 Report

Comments and Suggestions for Authors

The authors revised the article based on the first round review.

However, the authors should add efficiency for each primer to the table 1.

Moreover, the statistical section should be clear, but is not in the present form.

For ANOVA, the authors must state if normality and homoscdastisity was met or not. Also, it should be clarified how these assumptions were met or not.

The correct test for this experimental design is repeated measure two-way ANOVA. The authors may need to consult to a statistician.

Author Response

The authors revised the article based on the first round review.

However, the authors should add efficiency for each primer to the table 1.

Moreover, the statistical section should be clear, but is not in the present form.

For ANOVA, the authors must state if normality and homoscdastisity was met or not. Also, it should be clarified how these assumptions were met or not.

The correct test for this experimental design is repeated measure two-way ANOVA. The authors may need to consult to a statistician.

Response: Dear reviewer, thank you for pointing this out. We have added PCR efficiency for each primer to the table 1 and we have reanalyzed the data using the repeated measure two-way ANOVA.

Round 3

Reviewer 3 Report

Comments and Suggestions for Authors

.

Author Response

The correct test for this experimental design is repeated measure two-way ANOVA.

Response: Thanks for your advise. We have analyzed the data by repeated measure two-way ANOVA. And the result was also rewritten. I'm very sorry for causing you a bad feeling before.